# Adaptation to acidic conditions that mimic the tumor microenvironment, downregulates miR-193b-3p, and induces EMT via TGFβ2 in A549 cells

**Sadayuki Higashi, Munekazu Yamakuchi**®*, **Hirohito Hashinokuchi,**
**Kazunori Takenouchi, Akito Tabaru**®, **Yoko Oyama, Chieko Fujisaki, Kiyonori Tanoue,**
**Teruto Hashiguchi***

Department of Laboratory and Vascular Medicine, Graduate School of Medical and Dental Sciences,
Kagoshima University, Kagoshima, Japan

* k1581347@kadai.jp (TH); munekazu@m.kufm.kagoshima-u.ac.jp (MY)

org/10.1371/journal.pone.0318811

del Seguro Social, MEXICO

**Peer Review History:** PLOS recognizes the
benefits of transparency in the peer review
process; therefore, we enable the publication
of all of the content of peer review and
author responses alongside final, published
articles. The editorial history of this article is
available here: https://doi.org/10.1371/journal.
pone.0318811

## Abstract

The acidic tumor microenvironment plays a critical role in the malignant transformation of
cancer cells. One mechanism underlying this transformation involves epithelial-
mesenchymal transition (EMT). This is induced by prolonged exposure to acidic con-
ditions. EMT is an essential process in cancer progression, with Transforming Growth
Factor Beta (TGF-β) playing a central role in its induction. However, little was known
about the factors regulating TGF-β under acidic conditions. This study aimed to elucidate
the mechanism of EMT under acidic conditions and identify novel therapeutic targets
to inhibit cancer cell migration and metastasis. Focusing on lung cancer, we explored
microRNAs associated with EMT that were differentially expressed under acidic conditions
in A549 cells and identified miR-193b-3p as a novel candidate. Under acidic conditions,
miR-193b-3p expression decreased around days 3–14. Downregulation of miR-193b-3p
promoted increased TGFβ2 expression, resulting in EMT changes in A549 cells. Our
study suggests that the interaction between miR-193b-3p, TGFβ2, and the acidic tumor
microenvironment promotes cancer EMT change. Understanding these interactions may
not only enhance our biological comprehension of cancer, but also pave the way for the
development of targeted therapies to inhibit cancer metastasis.

## Introduction

An extracellular acidic environment, a characteristic of the tumor microenvironment, has been
reported to cause the malignant transformation of cancers such as colon, breast, and lung cancer
[1–3]. One of the causes of malignant transformation is thought to be epithelial-mesenchymal
transition (EMT) caused by long-term exposure to an acidic environment [1,4]. However, few
studies have investigated why EMT changes occur under acidic conditions [1].

　　EMT in cancer is an important biological process in tumor progression, and TGF-β plays
a central role in its induction [5,6]. EMT involves key transcription factors such as SNAIL,

**Data availability statement:** All relevant data are within the manuscript and its Supporting Information files.

**Funding:** This study was financially supported by the Japan Society for the Promotion Science (JSPS) (https://www.jsps.go.jp) in the form of a grant (22K16538) received by K. Tanoue. No additional external funding was received for this study.

**Competing interests:** The authors have declared that no competing interests exist.

TWIST, and ZEB, and upregulation of these transcription factors leads to the downregulation of the epithelial marker E-cadherin and an increase in the mesenchymal markers N-cadherin and vimentin [7–12]. Alterations in these molecules cause changes in cancer cell behavior and characteristics, contributing to morphological and functional changes, loss of epithelial cell characteristics, and mesenchymal cell behavior, becoming more motile in response to the surrounding environment. These changes promote migration, invasion, acquisition of resistance, and formation of cancer stem cells, which are important factors in tumor progression [13–17].

MicroRNAs (miRNAs) are non-coding RNAs of approximately 22 nucleotides that suppress the expression of target genes and perform various biological functions such as regulating development, differentiation, and tumor progression [18–20]. Especially in cancer, dysregulation of miRNAs contributes to carcinogenesis and tumor progression, as reported for miRNAs such as miR-17, miR-19, miR-21, miR-155, and miR-569 [21–26]. Although miRNAs are important regulators of malignant transformation in cancers, few studies have addressed the effects of changes in miRNAs on the acidic tumor microenvironment [27].

The aim of this study was to elucidate the mechanism of EMT under acidic conditions and to find a new treatment option by suppressing cancer cell migration and metastasis. In this study, on lung cancer, we searched for miRNAs associated with EMT that are differentially expressed under acidic conditions in A549 cells and identified miR-193b-3p as a new such miRNA. The results showed that miR-193b-3p expression was significantly decreased in cells cultured under acidic conditions. TGFβ2 was downregulated by miR-193b-3p, and decreased expression of miR-193b-3p resulted in increased TGFβ2 expression. We confirmed that these change in miR-193b-3p and TGFβ2 expression resulted in EMT changes in A549 cells under acidic conditions.

## Materials and methods

### Reagents

The reagents used for cell culture in this study were as follows: HEPES (Dojindo, Japan), Dulbecco's Modified Eagle's medium (DMEM; Cat# D5030, Sigma-Aldrich, MO, USA), glucose, $NaHCO_3$, and L-alanyl-L-glutamine (Fujifilm, Japan). The antibodies used in this study are listed in S1 Table Reagents used for cell function analysis are as follows: SB431542 (Cell Signaling Technology, MA, USA), recombinant human-TGFβ2 (R&D system, MN, USA).

### Cell culture

The human lung adenocarcinoma cell line A549 was obtained from ATCC and maintained at 37°C under 5% CO2 in DMEM containing 10% fetal bovine serum, 10mM HEPES, 0.1% $NaHCO_3$, 4mM L-alanyl-L-glutamine and 10 mM glucose. The pH of the media was measured using a pH meter (Horiba, Japan) and adjusted to pH 7.4 or pH 6.8 with 4N NaOH. Cells were cultured at pH 6.8 for up to 12 weeks, and the cells cultured at pH 7.4 were cultured for the same period. SB431542 (final conc. 10 μM) was treated with miR-193b inhibitor for 6 days prior to cell harvest.

### Transfection

The cells were transfected with Lipofectamine RNAiMAX (Thermo Fisher Scientific). Overexpression of miRNAs was performed using a 10 nM miRNA mimic (ID:MC12383, Thermo Fisher Scientific). miRNA inhibition experiments were performed using a 20 nM miRNA inhibitor (ID:MH12383, Thermo Fisher Scientific). As their controls, mirVana™miRNA

Mimic, negative control (ID:4464058, Thermo Fisher Scientific) and mirVana™miRNA inhibitor negative control (ID:4464076, Thermo Fisher Scientific) were used.

## Wound healing assay

A549 cells were cultured in 12-well plates. After 24 h of starvation in media containing 2% FBS, a scratch was made in the cell area using a 200µl pipette tip. The treated cells were washed with PBS. The cells that migrated into the scratched area were photographed at ×40 using an inverted microscope (bz810, Keyence, Japan). Cell migration data are expressed as a percentage of the migration rate after 48 or 72 h relative to the scratch distance at 0 h.

## Cell proliferation assay

Cells ($1 \times 10^4$ cells per well) were seeded in a 12-well plate and cultured for 3 days, then washed with PBS. Cells were treated with 100 µl per well of 0.05% trypsin/EDTA, and the detached cells were counted with a cell counter.

## Western blot assay

Cells were lysed in a cell lysis buffer (Cell Signaling Technology). Proteins were separated on a 4–15% SDS-PAGE gradient gel, transferred to a nitrocellulose membrane (Bio-Rad, Hercules, CA, USA), and blocked with 5% nonfat dried milk in PBST. The membranes were incubated overnight at 4°C with a primary antibody followed by 1 h incubation with an HRP-conjugated secondary antibody. The bands were visualized with SuperSignal™ Western Blot Enhancer (Thermo Fisher Scientific) and Amersham™ IQ500 (Cytiva, MA, USA). The data were quantified using Image J and normalized to the expression level of GAPDH.

## Immunofluorescence

Cells grown in glass-bottomed dishes were fixed with 4% paraformaldehyde (Fujifilm) for 30 min. After three washes with PBS, the cells were incubated with Alexa-555 phalloidin (Cell Signaling Technology) for 10 min, and the nuclei were counterstained with DAPI (Thermo Fisher Scientific). Finally, the cells were imaged using a microscope (LSM700, OLYMPUS, Japan).

## Microarray

Total RNA was obtained from A549 cells cultured at pH 7.4 or pH 6.8 for 4 weeks using 3D-Gene® RNA extraction reagent (Toray Industries, Kanagawa, Japan). The RNA samples were labeled using the 3D-Gene miRNA labeling kit and hybridized to Human miRNA Oligo chips. Images were obtained by scanning the microarray and the fluorescence signals were extracted by the 3D-GeneH Scanner 3000 (Toray Industries). The fluorescence intensities were normalized using global normalization across all microarray data.

## Real time PCR (RT-PCR)

Total RNA from cells was purified using the miRNeasy Mini Kit (Qiagen). RT-PCR was performed using reagents from Applied Biosystems (Foster City, CA, USA) according to the supplier's recommendations. Primers used are listed in S2 Table. The data were normalized to the expression level of GAPDH for mRNAs and U6 for miRNAs.

## Statistical analysis

Data are reported as mean ± standard error. Comparisons between two groups were analyzed using Student's t-test. Multiple comparisons were performed by one-way ANOVA with

Tukey's test. A p values less than 0.05 indicates a statistically significant population change. Asterisks in figures indicate the magnitude of the p value (*p < 0.05, **p < 0.01, *** p < 0.001); ns stands for no statistical significance.

## Results

### Culture under acidic conditions induces EMT-like changes in A549 cells

To examine the effects of an acidic microenvironment on cancer cells, we first established a culture system in a medium mimicking an acidic tumor environment (pH 6.8), which mimics the pH of cancer tissue, as previously reported [28,29]. Cells from the lung adenocarcinoma cell line A549 were cultured for up to 12 weeks. A549 cells cultured in an acidic environment for more than 3 weeks showed changes in cell morphology, with more spindle-shaped and protruding cells (Fig 1A). No significant change in the cell proliferation rate was observed after 3 days of incubation in an acidic environment (Fig 1B). Cultivation under acidic conditions caused an increase in the expression of mesenchymal markers (N-cadherin, vimentin) and a decrease in the expression of epithelial markers(E-cadherin) (Fig 1C). Furthermore, the migration capacity of A549 cells cultured under acidic conditions was enhanced, as was their invasive activity (Fig 1D and S1 Fig), suggesting that cultivation under acidic conditions resulted in EMT changes.

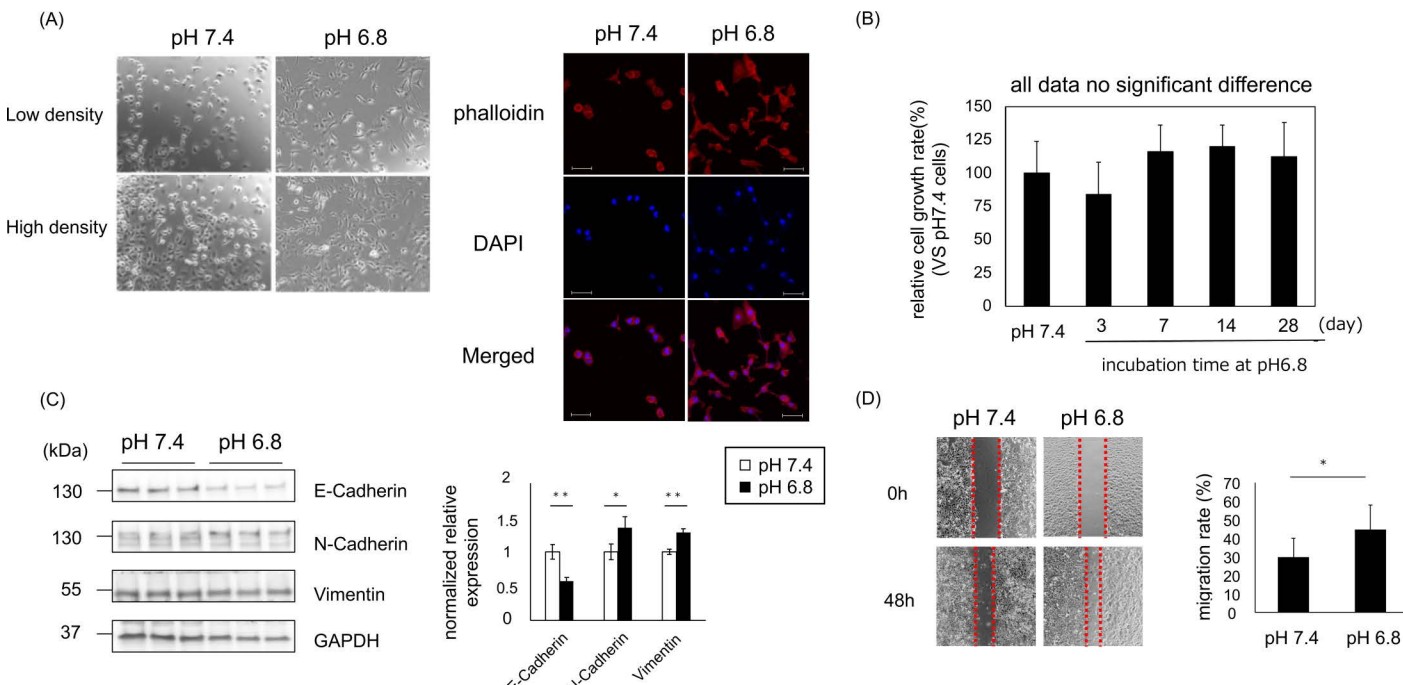

**Fig 1. Culture under acidic conditions induces EMT-like changes in A549 cells.** (A) Representative microphotographs and immunofluorescent pictures of pH 7.4 and pH 6.8 adapted A549 cells. (Red, phalloidin; Blue, DAPI). Bar = 50μm (B) Cell growth rate 3 days after incubation at pH 6.8 (day 3, day 7, day 14, and day 28) (C) Expression levels of the epithelial marker E-cadherin and the mesenchymal markers (N-cadherin, and vimentin) in pH 7.4 and pH 6.8 adapted A549 cells. (Left: representative blots; Right: quantification of the relative expression of proteins) (D) Migration rate of A549 cells adapted to pH 7.4 and pH 6.8 conditions. (Left: images taken at 0 and 48 h; Right: quantification of the migration rate). Values are expressed as mean ± SEM. NS, not significant; *P < 0.05, **P < 0.01, ***P < 0.001 compared with their respective counterparts. (All data except Fig 1B: n = 3; Fig 1B: n = 4).

## Acidic environment induces EMT in A549 cells in a time-dependent manner via TGFβ

Although chronic exposure to acidic conditions causes EMT changes in several cancer types [1,4,30], Detailed reports are scarce regarding the time needed for full EMT changes, including the decrease in E-cadherin and the increase in N-cadherin, to take place under acidic environments. We monitored the changes in EMT markers in A549 cells cultured under acidic conditions using western blotting. The results showed that only the mesenchymal markers, such as N-cadherin and vimentin, increased during short-term acidic conditions from day 3 to 14, whereas the epithelial marker E-cadherin decreased after 28 days of culture (Fig 2A). The expression of TGFβ2 in A549 cells cultured at pH 6.8 was already 2-fold enhanced at day 3 compared to cells cultured at pH 7.4, but it continued to increase and reached more than 3.5-fold at day 28. On the other hand, TGFβ1 expression was 1.5-fold on day 3 of the acidic culture, but did not increase further until day 28 (Fig 2B). While TGFβ2 resulted in EMT changes in A549 cells, low doses (0.04 ~ 0.4 ng/mL) of TGFβ2 were less likely to cause a decrease in the epithelial marker E-cadherin (Fig 2C). Also under acidic conditions, EMT induction was blocked by a TGFβ RI inhibitor (SB431542) (Fig 2D). These data suggest that TGFβ2 plays a central role in EMT induction under acidic conditions, which occurs in a stepwise manner.

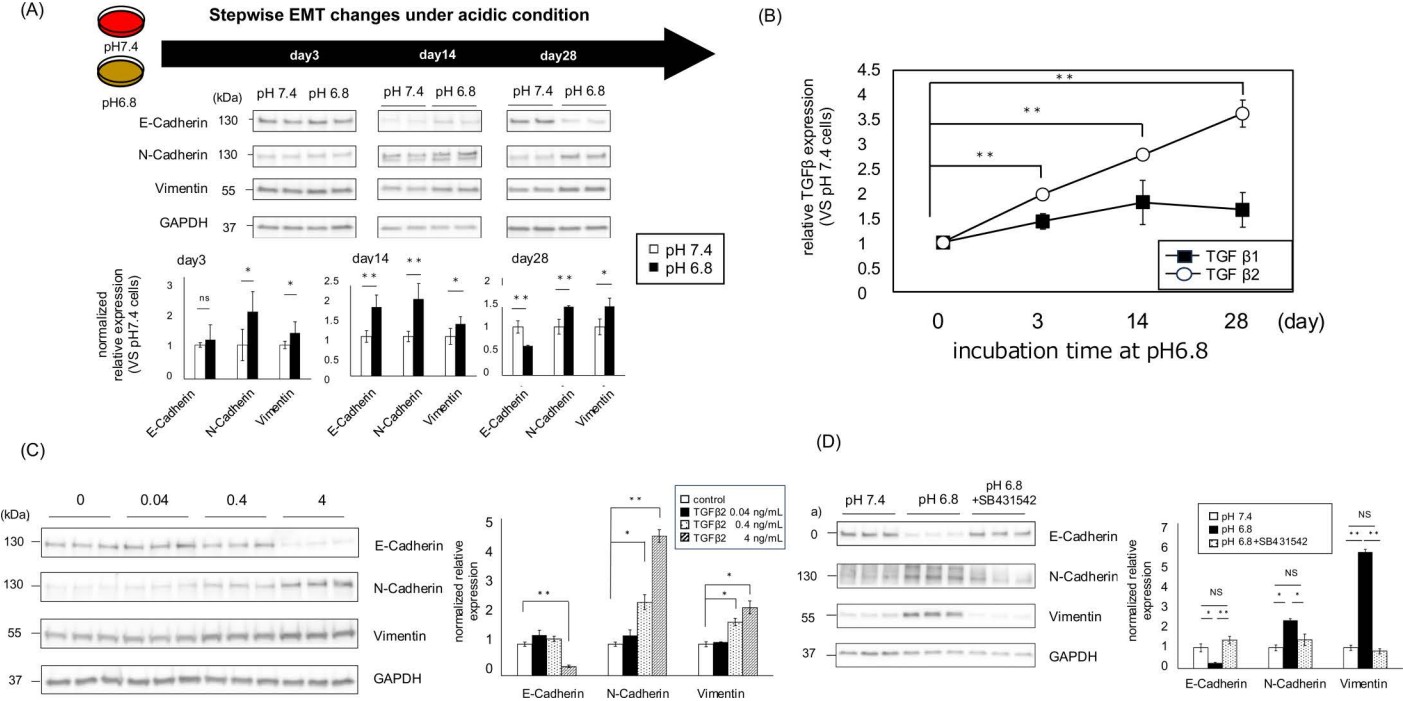

**Fig 2. TGF β2 expression and EMT induction in acidic conditioned cultures were continuously induced in a time-dependent manner in A549 cells.** (A) Stepwise EMT changes in cultures under acidic conditions. Representative immunoblotting for the epithelial marker E-cadherin and for the mesenchymal markers (N-cadherin and vimentin) after incubation at pH 6.8 (day 3, day 14, and day 28). (upper: representative blots; lower: quantification of the relative expression of proteins) (B) Time-dependent mRNA expression changes for TGFβ1 and TGFβ2 after incubation at pH 6.8 (day 3, day 14, and day 28) (C) The expression levels of the epithelial marker E-cadherin and the mesenchymal markers (N-cadherin and vimentin) with or without treatment with each concentration of TGFβ2 for 48 h. (Left: representative blots; Right: quantification of the relative protein expression) (D) EMT change induced after 4 weeks in an acidic environment was blocked by 10μM of a TGFβRl inhibitor (SB431542). Values are expressed as mean ± SEM. NS, not significant; *P < 0.05, **P < 0.01, ***P < 0.001 compared with their respective counterparts. (All data except Fig 1A: n = 3. Fig 1A: n = 2).

## In A549 cells, miR-193b-3p is downregulated in a time-dependent manner under acidic conditions

MicroRNA deregulation is involved in malignancy in cancers including EMT [20]. To elucidate whether changes in miRNA expression regulate the EMT under acidic conditions, we performed a comprehensive microarray analysis of miRNAs. Culturing under acidic conditions dramatically altered the miRNA expression profiles (Fig 3A). The results showed that ten miRNAs were upregulated by more than 2-fold, and only two miRNAs were downregulated by less than half. We focused on miR-193b-3p because it was most strongly down-regulated in acidic conditions (Fig 3A, 3B), and miR-193b has been reported to target TGFβ2 using the reporter gene assay in ATDC cells and HEK293 cells [31,32]. Using RT-PCR, we confirmed that miR-193b-3p expression was downregulated in a time-dependent manner under acidic conditions (Fig 3C). Interestingly, miR-193b-3p was not downregulated early (~3 d) during incubation under acidic conditions but was downregulated by day 14. This is consistent with the timing of EMT induction under acidic conditions (Fig 2A). Taken together, we hypothesized that downregulation of miR-193b-3p under acidic conditions modulates TGFβ2 expression and induces EMT in A549 cells.

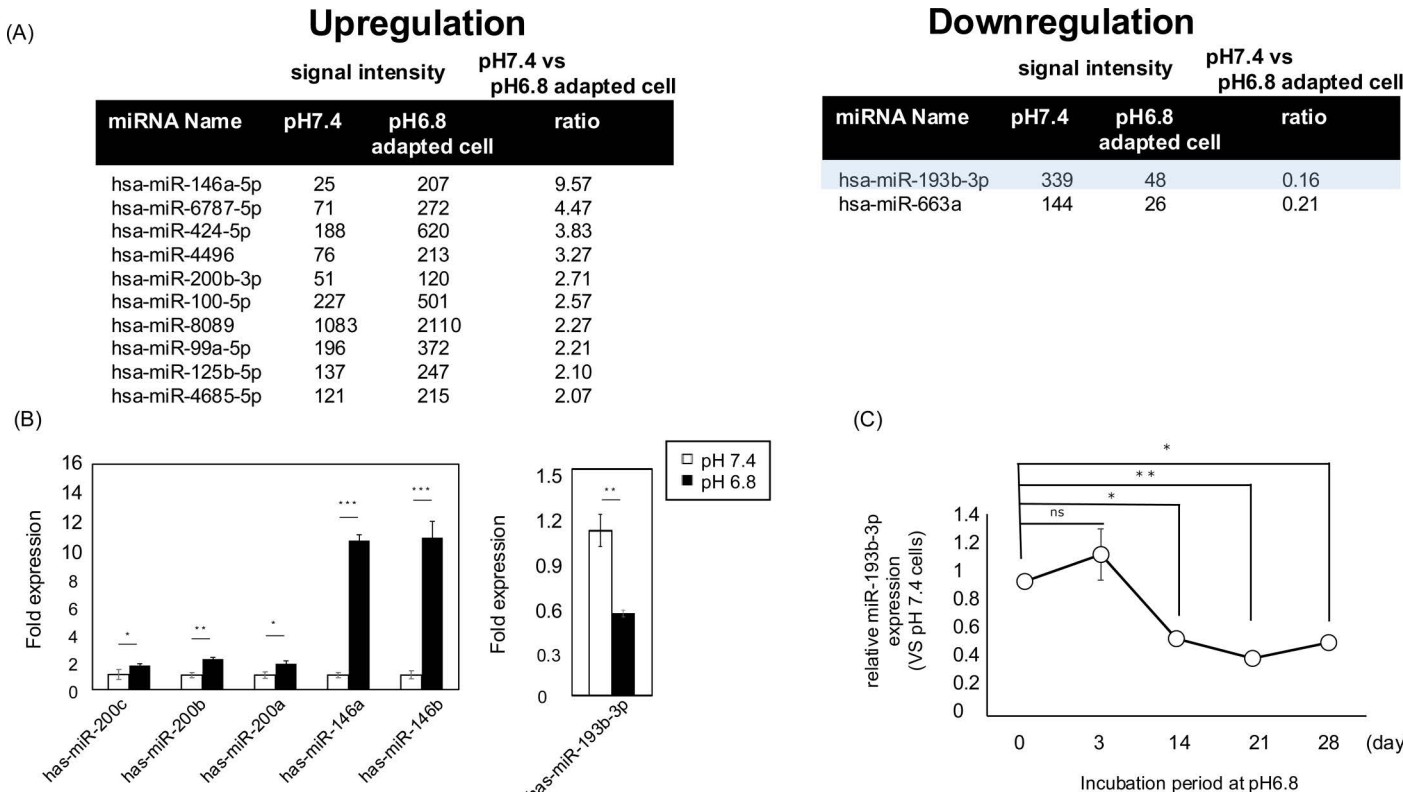

**Fig 3. Culture under acidic conditions downregulated miR-193b-3p in A549 cells.** (A) Microarray analysis (the list shows miRNAs that varied more than (left) or less than (right) 2-fold in culture under acidic conditions). The fluorescence intensities were normalized using global normalization across all microarray data. The "ratio" represents the relative expression levels of microRNAs in pH 6.8-adapted A549 cells relative to pH 7.4, calculated using Log2-transformed 75th percentile normalized values. The cut-off values were defined as a ratio > 2 for upregulated miRNAs and a ratio < 0.5 for downregulated miRNAs (B) Changes in miRNA expression under acidic conditions were evaluated using quantitative reverse transcription polymerase chain reaction (C) Time-dependent expression changes of miR-193b in acidic condition culture after incubation at pH 6.8 (day 3, day 14, day 21, and day 28). The graph shows the expression level of miR-193b-3p relative to that of A549 cells cultured at pH 7.4. Values are expressed as mean ± SEM. NS, not significant; *P < 0.05, **P < 0.01, ***P < 0.001 compared with their respective counterparts. (All data: n = 3).

## Downregulation of miR-193b-3p increases TGFβ2 expression and induces EMT in A549 cells under acidic conditions

To address our hypothesis, we investigated the relationship between miR-193b-3p and TGFβ. We transfected cells with the miR-193b-3p mimic and scrambled the control oligo. Over-expression of miR-193b-3p inhibited the expression of TGFβ2, but not TGFβ1 (Fig 4A) in A549 cells, resulting in a decrease in the mesenchymal markers (N-cadherin and vimentin) and an increase in the epithelial marker (E-cadherin) (Fig 4C). We further confirmed that the inhibitor of miR-193b-3p upregulates the expression of TGFβ2, but not TGFβ1 in A549 cells (Fig 4B) and resulted in an increase in the mesenchymal markers (N-cadherin and vimentin) and a decrease in the epithelial marker (E-cadherin), which was partially recovered by SB4314542, a TGFβRI inhibitor (Fig 4D). These findings support the notion that miR-193b-3p suppresses EMT by targeting TGFβ2, consistent with previous reports [31,32]. Furthermore, we investigated the changes in the migration ability of the miR-193b-3p mimic or inhibitor. Overexpression of miR-193b-3p led to reduced cell movement, whereas its inhibition significantly enhanced cell migration (Fig 4E, 4F). Notably, the suppression of EMT due to miR-193b-3p overexpression was partially restored by the addition of recombinant TGFβ2

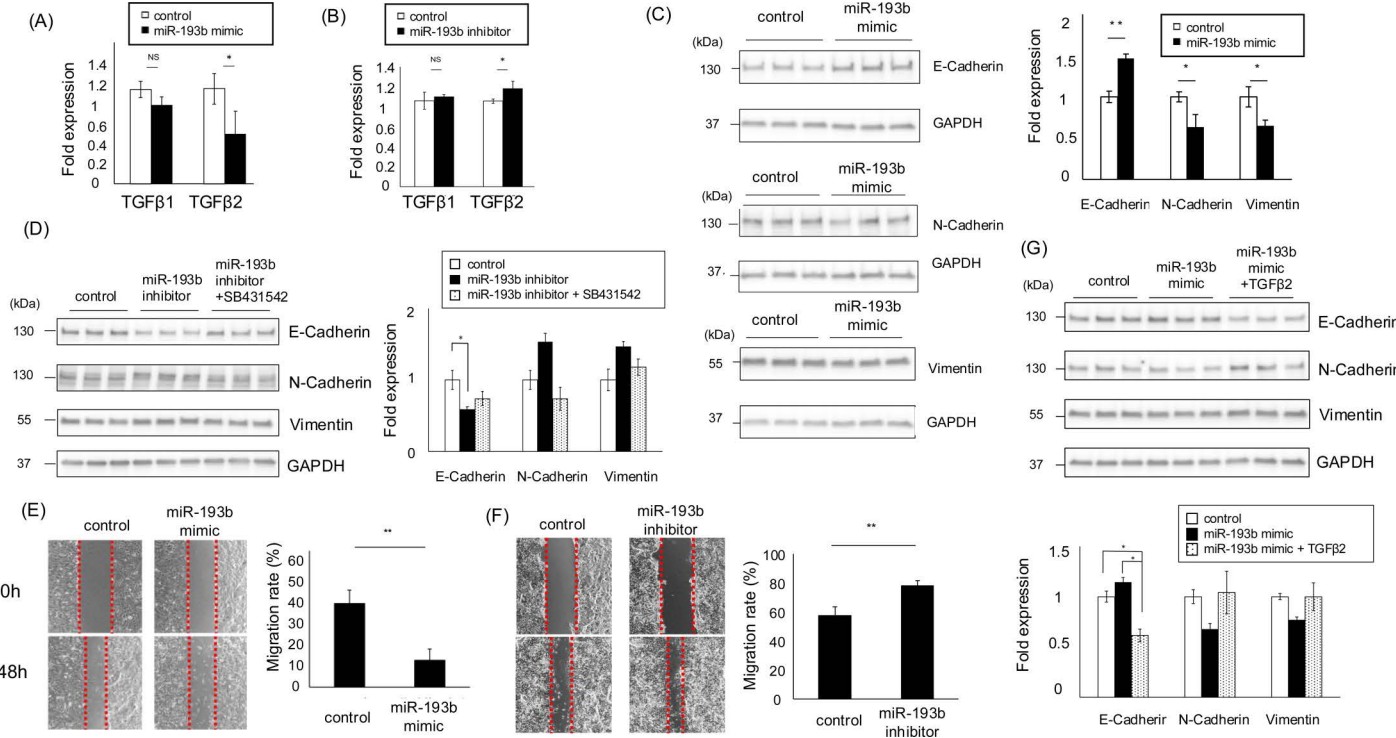

**Fig 4. miR-193b induces EMT change and migration ability through the regulation of TGFβ2 expression in A549 cells.** (A) miR-193b-3p overexpression inhibited the mRNA expression of TGFβ2 but not TGFβ1 in pH 6.8 adapted A549 cells (B) The miR-193 inhibitor enhanced the mRNA expression of TGFβ2 but not TGFβ1 mRNA expression in pH 7.4 adapted A549 cells (C) miR-193b-3p overexpressed, pH 6.8-adapted A549 cells showed an enhanced the epithelial marker (E-cadherin) and attenuated the mesenchymal markers (N-cadherin and vimentin). (Left: representative blots; Right: quantification of the relative protein expression) (D) miR-193b-3p knockdown in pH7.4 A549 cells attenuated the epithelial marker (E -cadherin) and enhanced the mesenchymal markers (N-cadherin, vimentin). (Left: representative blots; Right: quantification of the relative protein expression) (E) miR-193b overexpression in pH 6.8-adapted A549 cells increased the cell migration ability. (Left: images taken at 0 and 48 h; Right: quantification of migration rate) (F) A549 cells transfected with the miR-193b inhibitor showed reduced cell migration. (Left: images taken at 0 and 72 h; Right: migration rate quantification) (G) EMT marker changes induced by miR193b-3p overexpression were restored by the addition of TGFβ2. Values are expressed as mean ± SEM. NS, not significant; *P < 0.05, **P < 0.01, ***P < 0.001 compared with their respective counterparts. (All data: n = 3).

(1 ng/mL) (Fig 4G). Taken together, these results demonstrate that downregulation of miR-193b-3p elevates TGFβ2 expression, inducing EMT in A549 cells under acidic conditions.

## Discussion

Previous studies demonstrate that an acidic tumor microenvironment is a malignant cancer factor [3,33,34]. However, the underlying molecular mechanism remains unclear. We shed light on the role of miR-193b-3p and TGFβ2 in the acidic tumor microenvironment. We showed that downregulation of miR-193b-3p induces EMT and enhances TGFβ2 expression in A549 cells under acidic conditions. This suggests that acidic tumor microenvironment-induced changes in miRNAs and TGFβ2 may contribute to tumor progression and metastasis.

Various cancer microenvironments lead to the overexpression of TGFβ in many cancer tissues, including lung cancer, and plays an important role in tumor progression and metastasis [35,36]. The cancer microenvironment is composed of a variety of cells and components, including fibroblasts, immune cells, vascular cells, and the extracellular matrix, all of which interact to influence tumor growth and behavior [37]. These cells trigger local inflammation in the tumor microenvironment, resulting in hypoxic and acidic conditions around the tumor [37,38]. Hypoxia signaling is mainly mediated by HIFs (hypoxia-inducible factors), which induces TGFβ expression, promotes EMT, and stimulates tumor angiogenesis [39–41]. The involvement of TGFβ has also been suggested in an acidic environment [1,4].

It is extremely important to recognize that the EMT is a gradual and coordinated progression of several processes and that it takes a certain amount of time to complete the full EMT [13,42]. In the present study, it took 28 days for complete EMT changes (increased N-cadherin expression and decreased E-cadherin expression) to develop after exposure to an acidic environment. In A549 cells exposed to an acidic environment, there was an increase in N-cadherin and vimentin on day 3, followed by a decrease in E-cadherin on day 28 (Fig 2A). In the experiment with recombinant TGFβ2, increases in N-cadherin and vimentin were shown at lower concentrations, while E-cadherin did not decrease until higher concentrations (Fig 2C). Under acidic tumor microenvironment conditions, a marked increase in TGFβ2 compared to TGFβ1 was observed (Fig 2B). Although many studies have linked increased expression of total TGFβ to EMT [4], while others have explained the mechanism of EMT by TGFβ2 [1,43]. In clinical studies, TGFβ2 has been implicated in a variety of human malignancies. In gastric cancer, TGFβ2 correlated with poorer prognosis than TGFβ1 [44], and in malignant gliomas and nasopharyngeal carcinoma, TGFβ2 was associated with malignancy [45,46]. In addition, gallbladder cancer has shown an association between EMT and TGFβ2 in cancer metastasis [47], suggesting the possibility of specific actions of TGFβ2 in various cancers. TGFβ1 and TGFβ2 show subtle differences in structure, expression regulation and biological function [43,48,49], and the three TGFβ receptors (TGFBRI, TGFBRII and TGFBRIII) with different affinities [50] and thus different effects on EMT are expected. In addition, to our knowledge, there are no reports of increased expression of TGFβ3 in acidic tumor environments. Few studies have compared TGFβ1 and TGFβ2 in the same situation as in the present study. Future studies are expected to elucidate the differential expression and actions of these isoforms in various cancers.

The role of miR-193b-3p as a tumor suppressor has been documented in a variety of cancer types, including lung cancer [51–53]. miR-193b-3p may be involved in many biological processes, especially cancer progression, including the regulation of EMT, inhibition of cell proliferation, promotion of apoptosis, and cell cycle regulation [51,54–56] In this study, overexpression of miR-193b-3p in A549 cells reduced TGFβ2 mRNA levels by approximately

50%, whereas no significant change in TGFβ1 mRNA levels was observed (Fig 4A), indicating TGFβ2-specific regulation by miR-193b-3p. TGFβ2 has been reported to be one of target genes of miR-193b-3p, consistent with this finding [31,32]. Furthermore, the expression level of miR-193b-3p was unchanged immediately after exposure to acidic conditions (~3 days) but significantly decreased after 14 days, and these changes in miR-193b-3p expression levels were inversely correlated with changes in TGFβ2 expression and EMT (Figs 2A, 2B and 3C). These results suggest that acidic conditions initially induce low levels of TGFβ2 expression (days 3-14), with a further increase in TGFβ2 up to day 28, leading to a transition from partial to full EMT. In other words, under acidic conditions, the process of EMT proceeds in stages due to time-dependent changes in TGFβ2 expression, and the involvement of different molecules at each stage is expected.

This study has several limitations. First, this study was performed on a single cancer cell line. Whether similar results can be obtained with other lung cancer cells or cancer cells from other tissues should be investigated in the future. In addition, the in vitro cancer cell experiments do not fully reflect the in vivo tumor environment. In vivo experiments are needed since the actual tumor environment is formed by many cell-cell interactions as described above. Second, we did not examine the regulatory factors of miR-193b-3p in this study. It has been reported that histone acetylation is significantly altered during oxidative stress such as irradiation, resulting in downregulation of miR-193b [57]. There are some reports that histone acetylation is altered in tumor cells cultured under acidic conditions [58]. and that the addition of HDAC inhibitors to breast cancer cell lines blocked the expression of TGFβ2 [59]. Therefore, it is quite possible that epigenetic regulation is occurring in tumor cells under acidic conditions. Third, the effects of other miRNA changes must also be considered. Recently, regulation of TGFβ2 by miR-7 was reported in a non-small cell carcinoma cell line cultured under acidic conditions [60]. Our microarray analysis revealed downregulation of miR-7; however, the signal intensity was too low to be included in the analysis (data not shown). Although we found several miRNAs that were upregulated in acidic environments, the association between changes in these miRNAs and EMT has not yet been examined. For a more comprehensive understanding of the acidic tumor microenvironment, it is important to consider other miRNAs in addition to miR-193b-3p and analyze them in an integrated manner.

In conclusion, this study provides new insights into the interactions of miR-193b-3p, TGFβ2, and the acidic conditions such as tumor microenvironment. These results suggest that miR-193b-3p and TGFβ2 may be involved in the metastatic mechanism of tumors, at least *in vitro*, and further *in vitro* experiments using other tumor cells and *in vivo* experiments using animal models should be investigated.

## Supporting information

**S1 Table. Primary antibodies in this study.**
(PDF)

**S2 Table. Sequences of gene-specific primers for RT-PCR and primer ID in this study.**
(PDF)

**S1 Fig. Invasion assay.**
(PDF)

**S2 Fig. Western blot raw data.**
(PDF)

**S1 Raw Image. Raw Images of all Western blotting data.**
(PDF)

## Acknowledgments

The authors would like to thank Tomoka Nagasato and Akiko Katano (Department of Laboratory and Vascular Medicine) for their valuable contributions and Editage (www.editage.jp) for English language editing.

## Author contributions

**Conceptualization:** Sadayuki Higashi, Munekazu Yamakuchi, Teruto Hashiguchi.

**Data curation:** Sadayuki Higashi, Kiyonori Tanoue.

**Formal analysis:** Sadayuki Higashi.

**Funding acquisition:** Kiyonori Tanoue.

**Investigation:** Sadayuki Higashi, Hirohito Hashinokuchi, Kazunori Takenouchi, Akito Tabaru, Yoko Oyama, Chieko Fujisaki.

**Methodology:** Sadayuki Higashi.

**Supervision:** Munekazu Yamakuchi, Kiyonori Tanoue, Teruto Hashiguchi.

**Writing – original draft:** Sadayuki Higashi.

**Writing – review & editing:** Munekazu Yamakuchi, Teruto Hashiguchi.

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
