## [Decision Letter · Decision Letter 0]

12 Nov 2024

PONE-D-24-11976Adaptation to acidic conditions that mimicsthe tumor microenvironment, downregulates miR-193b-3p, and induces EMT via TGFβ2 in A549 cellsPLOS ONE

Dear Dr. Yamakuchi,

Thank you for submitting your manuscript to PLOS ONE. After careful consideration, we feel that it has merit but does not fully meet PLOS ONE’s publication criteria as it currently stands. Therefore, we invite you to submit a revised version of the manuscript that addresses the points raised during the review process.

We look forward to receiving your revised manuscript.

Kind regards,

Arturo Aguilar-Rojas

Academic Editor

PLOS ONE

2. Please amend either the title on the online submission form (via Edit Submission) or the title in the manuscript so that they are identical.

Additional Editor Comments:

According to the observations of reviewers 1 and 2, the article requires a thorough restructuring as well as an expansion of specific points in the discussion. Based on this, the authors are invited to review each point raised by the reviewers and respond to them. It is worth noting that, based on their responses and the restructuring made to the work, it will be re-evaluated.

Reviewers' comments:

Reviewer's Responses to Questions

**Comments to the Author**

1. Is the manuscript technically sound, and do the data support the conclusions?

Reviewer #1: Partly

Reviewer #2: Yes

2. Has the statistical analysis been performed appropriately and rigorously? 

Reviewer #1: No

Reviewer #2: Yes

3. Have the authors made all data underlying the findings in their manuscript fully available?

Reviewer #1: No

Reviewer #2: Yes

4. Is the manuscript presented in an intelligible fashion and written in standard English?

Reviewer #1: Yes

Reviewer #2: Yes

5. Review Comments to the Author

Reviewer #1: Minor points

• Abstract (line 30): wording is confusing, implies that miR-193b-3p downregulates TGFB2

• Figure 2A: why is the band for N-cadherin a doublet at Day 14, but not at other timepoints?

• When does TGFb2 increase with acidic culture conditions? There is already a 2-fold increase at day 3, the earliest timepoint shown. Understanding the onset of signaling changes in this model would be of interest to researchers seeking to build from this report. Similarly, there is a long gap between Day 3 and Day 14, that would benefit from resolution of when gene expression changes are first observed.

• Figure 1C and 2A appear to be partially redundant. Authors should show quantification of EMT protein expression changes for each of the timepoints studied.

• What is the duration of treatments tested in figure 4D?

Major points:

• Study should indicate which statistical tests were performed to establish significance in each figure. T-tests are only appropriate when two conditions are tested. Most comparisons in the study should be evaluated using One-way ANOVA or 2Way ANOVA (depending on the number of factors in the experiment), with a multiple comparisons test, such as Tukey, performed to measure if differences between pairs of means are significant.

• The authors implicate that under acidic culture conditions, loss of miR-193b-3p promotes EMT via upregulation of TGFB2. However figure 2B shows that TGFB2 expression is increased 2-fold by day 3, whereas figure 3C shows that miR-193b-3p levels are not reduced until day 14. The authors should seek to reconcile the timeline of expression changes under acidic culture conditions with their hypothesis. How much of acidity-induced changes in TGFB2 increase can be attributed to miR-193b-p?

• To connect the proposed pathway to acidity-induced EMT, the authors should evaluate modulation of miR-193b-3p in cells cultured at ph 6.8 vs ph 7.4.

• The implications of this study are significantly limited by evaluation of a single cancer cell line. Evaluation of additional cell lines and/or primary tumor cells would broaden the potential impact of this work by highlighting whether these phenomena are common across different lines. Given the variability between different cancer lines, the current study's interpretations may potentially only be applicable to A549 cells.

Reviewer #2: The manuscript focuses on a significant topic in cancer biology: the impact of an acidic tumor microenvironment on epithelial-mesenchymal transition (EMT), with a specific focus on miR-193b-3p and TGFβ2 in lung adenocarcinoma (A549) cells. The study presents novel insights and a potential therapeutic target, which are of substantial interest to the scientific community. However, the following observations/additions may be considered to improve this very well written manuscript:

Major concerns

1. While the manuscript convincingly shows miR-193b-3p downregulation under acidic conditions leading to EMT, the mechanistic pathway between the acidic environment and miR-193b-3p downregulation is not fully elucidated. What is the possible mechanism for the acidic environment induce miR-193b-3p downregulation? A more detailed exploration of upstream regulatory factors would strengthen the findings and could enrich the discussion.

2. The study highlights the role of TGFβ2 but briefly discusses TGFβ1 without deep comparative analysis. Clarifying why TGFβ2 plays a dominant role under acidic conditions and how it diverges functionally from TGFβ1 would enhance the discussion.

3. While the wound healing assay is a valuable tool, employing other methodologies, such as the Transwell migration assay and animal models, would provide a more comprehensive understanding of cancer cell dynamics and EMT process.

4. Is there a canonical miR-193b-3p binding site in the 3'UTR of the TGFβ2 gene? It would be great if the authors could confirm that TGFβ2 gene is a target of miR-193b-3p through Luciferase Reporter Assay.

5. The conclusions drawn are broadly consistent with the data. The manuscript demonstrates that miR-193b-3p suppression in acidic environments leads to EMT through TGFβ2 upregulation, contributing to cancer cell motility and progression. However, the authors should temper their claims about potential therapeutic applications, given the study’s in vitro nature, but could expand on how the in vitro findings translate to clinical settings or animal models in discussion section.

Minor concerns

1. In methods, the authors should clarify the method used to normalize the fluorescence intensities of the microarrays, as well as the definition of the “ratio” and the cut-off value to determine upregulated and downregulated miRNAS.

2. In line 262, the authors first describe the results of cell migration in cells transfected with the mimic and then describe the results in cells transfected with the inhibitor. However, Figure 4 panel E shows the results of the effects of the inhibitor on cell migration and panel F shows the results of the effects of the mimic on cell migration. Authors should submit figures in the order described in the text.

The manuscript has substantial merit and could be a valuable contribution to the field after addressing the above concerns. I recommend minor to moderate revisions to improve clarity, robustness, and depth in interpretation, particularly regarding the mechanistic pathways and limitations of the study.

6. PLOS authors have the option to publish the peer review history of their article (what does this mean? ). If published, this will include your full peer review and any attached files.

**Do you want your identity to be public for this peer review?** For information about this choice, including consent withdrawal, please see our Privacy Policy .

Reviewer #1: No

Reviewer #2: **Yes: ** PhD. Mauricio Flores-Fortis

---

## [Author Response · Author response to Decision Letter 1]

28 Dec 2024

Author Response to Reviewer Comments

We have carefully considered each reviewer’s comment and have addressed each comment below. Specific revisions made to the manuscript text are flagged using Track Changes.

Response to Reviewer #1

We appreciate the Reviewer #1 comments. We have carefully read through all the concerns and corrected or modified the manuscript as addressed below.

Minor points

• Abstract (line 30): wording is confusing, implies that miR-193b-3p downregulates TGFB2

Author response

Thank you for your comment. We have rewritten the sentences you pointed out (Page 2, Lines 28-31) to the following new sentence.

“Under acidic conditions, miR-193b-3p expression decreased around Days 3-14. Downregulation of miR-193b-3p promoted increased TGFβ2 expression, resulting in EMT changes in A549 cells.”

• Figure 2A: why is the band for N-cadherin a doublet at Day 14, but not at other timepoints?

Author response

The second band is most likely a degradation or glycosylation product of N-cadherin. The antibody we used (#13116 Cell Signaling Technology) also yielded a band that is a doublet. Our results of Fig 2A show a similar trend for both doublet bands on Day 14. The significance of why the two bands are observed only on day 14 is unclear, however, we believe the interpretation of the results would be the same for either N-cadherin band.

• When does TGFb2 increase with acidic culture conditions? There is already a 2-fold increase at day 3, the earliest timepoint shown. Understanding the onset of signaling changes in this model would be of interest to researchers seeking to build from this report. Similarly, there is a long gap between Day 3 and Day 14, that would benefit from resolution of when gene expression changes are first observed.

Author response

We appreciate these important comments. Since the purpose of this study was to examine morphological and miRNAs' changes in long-term cultures (4 weeks), short-term cultures were measured for only two points, day 3 and day 14. Therefore, the dynamics of TGFb have not been fully validated. We concluded that the decreased expression of miR-193b in an acidic environment after day 14 resulted in the increased expression of TGFb2 observed on day 28. As reviewer 1 pointed out, the change over time in short-term culture is a very important perspective, and perhaps the early (1-2 weeks) increase in TGFb2 expression may involve miRNAs other than miR-193b. We plan to study in detail the changes in TGFβ2 expression in the early stages of the acidic environment of this process. Based on your comments, we have revised the text in Result (Page 12, Lines 192-196) as follows.

"The expression of TGFβ2 in A549 cells cultured at pH 6.8 was already 2-fold enhanced at day 3 compared to cells cultured at pH 7.4, but it continued to increase and reached more than 3.5-fold at day 28. On the other hand, TGFβ1 expression was 1.5-fold on day 3 of the acidic culture, but did not increase further until day 28."

• Figure 1C and 2A appear to be partially redundant. Authors should show quantification of EMT protein expression changes for each of the timepoints studied.

Author response

Thank you for your suggestion. We changed the blotting picture into the new one and quantified it in Figure 1C (New Figure 1C). We also quantified EMT protein expressions in Western blotting and created bar graphs (New Figure 2A) and changed its caption.

• What is the duration of treatments tested in figure 4D?

Author response

The experiment was performed with A549 cells adapted to pH 6.8 for more than 8 weeks. After treatment with miR-193b inhibitor and/or SB431542 for 6 days, these cells were used for further experiments. The following sentences were added in the methods section (Page 5, Lines 87-88) and we changed the caption in Figure 4D.

“SB431542 (final conc. 10 ��) was treated with miR-193b inhibitor for 3 days prior to cell harvest.“

Major points:

• Study should indicate which statistical tests were performed to establish significance in each figure. T-tests are only appropriate when two conditions are tested. Most comparisons in the study should be evaluated using One-way ANOVA or 2Way ANOVA (depending on the number of factors in the experiment), with a multiple comparisons test, such as Tukey, performed to measure if differences between pairs of means are significant.

Author response

We appreciate your valuable comments. We checked all the statistical data and changed or modified them (Fig 1B, 2A, 2B, 2C, 2D, 3C, 4D, 4G). We also rewrote the statistical analysis section in Material and Method (Page 9, Lines 145-149) as follows. The respective statistical results are reflected in the text.

“Data are reported as mean ± standard error. Comparisons between two groups were analyzed using Student’s t-test. Multiple comparisons were performed by one-way ANOVA with Tukey’s test. A p values less than 0.05 indicates a statistically significant population change. Asterisks in figures indicate the magnitude of the p value (*p < 0.05, **p < 0.01, *** p < 0.001); ns stands for no statistical significance.”

• The authors implicate that under acidic culture conditions, loss of miR-193b-3p promotes EMT via upregulation of TGFB2. However figure 2B shows that TGFB2 expression is increased 2-fold by day 3, whereas figure 3C shows that miR-193b-3p levels are not reduced until day 14. The authors should seek to reconcile the timeline of expression changes under acidic culture conditions with their hypothesis. How much of acidity-induced changes in TGFB2 increase can be attributed to miR-193b-p?

Author response

As noted by reviewer #1, downregulation of miR-193b-3p is observed from day 14 (Figure 3C). Therefore, it is likely that regulators other than miR-193b-3p are involved in the increase in TGFβ2 on day 3. In other words, a group of genes responsive to the acidic environment may have been expressed and rapidly increased TGFβ2 expression. However, EMT only partially occurred at day 3 and 14, and TGFβ2 expression further increased and EMT continued to progress through day 28. This indicates that TGFβ2 expression does indeed increase until day 14 but may not be secreted in sufficient amounts to complete EMT, suggesting that regulation of TGFβ2 by miR-193b-3p is essential for completion of EMT.

We rewrote the section describing this point (Pages 21-22, Lines 353-358).

“These results suggest that acidic conditions initially induce low levels of TGFβ2 expression (days 3-14), with a further increase in TGFβ2 up to day 28, leading to a transition from partial to full EMT. In other words, under acidic conditions, the process of EMT proceeds in stages due to time-dependent changes in TGFβ2 expression, and the involvement of different molecules at each stage is expected.”

• To connect the proposed pathway to acidity-induced EMT, the authors should evaluate modulation of miR-193b-3p in cells cultured at ph 6.8 vs ph 7.4.

Author response

Although we did not examine the regulatory factors of miR-193b-3p in this study, it has been reported that histone acetylation is significantly altered during oxidative stress such as irradiation, resulting in downregulation of miR-193b (Ref. #57). There are some reports that histone acetylation is altered in tumor cells cultured under acidic conditions (Ref. #58) and that the addition of HDAC inhibitors to breast cancer cell lines blocked the expression of TGFβ2 (Ref. #59). Therefore, it is quite possible that epigenetic regulation is occurring in tumor cells under acidic conditions, and although not tested for miRNAs in references 2 and 3, these epigenetic changes may be responsible for the downward regulation of miR-193b-3p observed in this study.

*References

Ref. #57）Low-dose irradiation promotes Rad51 expression by down-regulating miR-193b-3p in hepatocytes. Sci Rep. 2016 May 26;6:25723.

Ref. #58）Tumor-specific metabolic adaptation to acidosis is coupled to epigenetic stability in osteosarcoma cells. Am J Cancer Res. 2016 Mar 15;6(4):859-75.

Ref. #59）HDAC inhibitors induce LIFR expression and promote a dormancy phenotype in breast cancer. Oncogene. 2021 Aug;40(34):5314-5326.

Based on these considerations, the following sentences were added to the Discussion section (Page 22, Lines 364-370) and remade this paragraph.

“Second, we did not examine the regulatory factors of miR-193b-3p in this study. It has been reported that histone acetylation is significantly altered during oxidative stress such as irradiation, resulting in downregulation of miR-193b (Ref. #57). There are some reports that histone acetylation is altered in tumor cells cultured under acidic conditions (Ref. #58) and that the addition of HDAC inhibitors to breast cancer cell lines blocked the expression of TGFβ2 (Ref. #59). Therefore, it is quite possible that epigenetic regulation is occurring in tumor cells under acidic conditions.”

• The implications of this study are significantly limited by evaluation of a single cancer cell line. Evaluation of additional cell lines and/or primary tumor cells would broaden the potential impact of this work by highlighting whether these phenomena are common across different lines. Given the variability between different cancer lines, the current study's interpretations may potentially only be applicable to A549 cells.

Author response

Thank you for your valuable comment. As you pointed out, this study used a single cell line of A549 cells, therefore, our concept is not applicable to general cancer biology. However, we have found that the micro-acidic environment of cancer cells is linked to EMT via specific miRNA, miR-193b-5p. We plan to investigate whether the EMT regulation by miR-193b-5p, which we have confirmed in a single cancer cell line, can be generalized. We added the following sentences in the Discussion section (Page 22, Lines 359-364).

" First, this study was performed on a single cancer cell line. Whether similar results can be obtained with other lung cancer cells or cancer cells from other tissues should be investigated in the future. In addition, the in vitro cancer cell experiments do not fully reflect the in vivo tumor environment. In vivo experiments are needed since the actual tumor environment is formed by many cell-cell interactions as described above.”

Response to Reviewer #2

We appreciate Reviewer #2 comments. Reviewer #2 stated “The study presents novel insights and a potential therapeutic target, which are of substantial interest to the scientific community” and raised important comments. We have carefully read through all the concerns and corrected or modified the manuscript as addressed below.

Major concerns

1. While the manuscript convincingly shows miR-193b-3p downregulation under acidic conditions leading to EMT, the mechanistic pathway between the acidic environment and miR-193b-3p downregulation is not fully elucidated. What is the possible mechanism for the acidic environment induce miR-193b-3p downregulation? A more detailed exploration of upstream regulatory factors would strengthen the findings and could enrich the discussion.

Author response

Thank you for pointing out an important issue in our manuscript. In this study, we did not examine the regulatory factors of miR-193b-3p because we aim to identify a miRNA that regulates EMT under acidic conditions in the cancer microenvironment. Oxidative stress caused by radiation has been shown to alter histone acetylation and to decrease miR-193b expression (53). Acidic conditions also change the level of histone acetylation in tumor cells (54,55). Therefore, transcriptional changes by acetylation of histone might act on the expression of miR-193b in tumors observed in this study.

A text about these has been inserted in Discussion (Page 22, Lines 364-370).

*References

53. Yu H, Peng Y, Wu Z, Wang M, Jiang X. MiR-193b as an effective biomarker in human cancer prognosis for Asian patients: a meta-analysis. Transl Cancer Res. 2022;11: 2249–2261.

54. Jian B, Li Z, Xiao D, He G, Bai L, Yang Q. Downregulation of microRNA-193-3p inhibits tumor proliferation migration and chemoresistance in human gastric cancer by regulating PTEN gene. Tumour Biol. 2016;37: 8941–8949.

55. Li C, Chen Y, Chen X, Wei Q, Cao B, Shang H. Downregulation of MicroRNA-193b-3p Promotes Autophagy and Cell Survival by Targeting TSC1/mTOR Signaling in NSC-34 Cells. Front Mol Neurosci. 2017;10: 160.

2. The study highlights the role of TGFβ2 but briefly discusses TGFβ1 without deep comparative analysis. Clarifying why TGFβ2 plays a dominant role under acidic conditions and how it diverges functionally from TGFβ1 would enhance the discussion.

Author response

Thank you for your comment. Since TGFβ1 expression changes little in long-term culture in acidic environments and TGFβ2 is generally considered to be more strongly involved in EMT than TGFβ1 in cancer cells, we focused this study on the relationship between TGFβ2 and EMT. We rewrote the description of the importance and specificity of TGFβ2 in EMT in the Discussion section as follows (Pages 20-21, Lines 326-341).

“Under acidic tumor microenvironment conditions, a marked increase in TGFβ2 compared to TGFβ1 was observed (Figure 2B). Although many studies have linked increased expression of total TGFβ to EMT [4], while others have explained the mechanism of EMT by TGFβ2 [1,46]. In clinical studies, TGF-b2 has been implicated in a variety of human malignancies. In gastric cancer, TGFβ2 correlated with poorer prognosis than TGFβ1 [44], and in malignant gliomas and nasopharyngeal carcinoma, TGFβ2 was associated with malignancy [45,46]. In addition, gallbladder cancer has shown an association between EMT and TGFβ2 in cancer metastasis [47], suggesting the possibility of specific actions of TGFβ2 in various cancers. TGFβ1 and TGFβ2 show subtle differences in structure, expression regulation and biological function [43, 48, 49], and the three TGFβ receptors (TGFBRI, TGFBR2 and TGFBRIII) with different affinities [50] and thus different effects on EMT are expected. In addition, to our knowledge, there are no reports of increased expression of TGFβ3 in acidic tumor environments. Few studies have compared TGFβ1 and TGFβ2 in the same situation as in the present study. Future studies are expected to elucidate the differential expression and actions of these isoforms in various cancers.”

3. While the wound healing assay is a valuable tool, employing other methodologies, such as the Transwell migration assay and animal models, would provide a more comprehensive understanding of cancer cell dynamics and EMT process.

Author response

Thank you for your feedback. In addition to the wound healing assay, we performed an invasion assay comparing pH 7.4 and pH 6.8-adapted A549 cells. Although it is limited to this comparison, we have included it as a S1 Fig and added the following sentence in Results section (Page 10, Line 164-166).

“ Furthermore, the migration capacity of A549 cells cultured under acidic conditions was enhanced, as was their invasive activity (Fig 1D and S1 Fig).”

4. Is there a canonical miR-193b-3p binding site in the 3'UTR of the TGFβ2 gene? It would be great if the authors could confirm that TGFβ2 gene is a target of miR-193b-3p through Luciferase Reporter Assay.

Author response

We appreciate your suggestion. TGF-�2 has already been shown to be one of miR-193b target genes in HEK293 cells (ref. 4) and ATDC cells (ref. 5) by using Luciferase Reporter Assay. We agreed with the reviewer #2 that adding Luciferase Reporter Assay would be even better. However, instead of doing this experiment, we added the following sentence and cited these two papers as evidence that miR-193b directly regulates TGF-�2 expression (Page 14, Line 230-232).

“…, miR-193b has been reported to target TGF��� using the reporter gene assay in ATDC cells and HEK293 cells [31,32].”

*References

Ref. 31) Zhou X, et al. The aberrantly expressed miR-193b-3p contributes to preeclampsia through regulating transforming growth factor-β signaling. Sci Rep. 2016 Jan 29;6:19910.

Ref. 32) Hou C, et al. MiR-193b regulates early chondrogenesis by inhibiting the TGF-beta2 signaling pathway. FEBS Lett. 2015 Apr 13;589(9):1040-7.

5. The conclusions drawn are broadly consistent with

---

## [Decision Letter · Decision Letter 1]

22 Jan 2025

Adaptation to acidic conditions that mimic the tumor microenvironment, downregulates miR-193b-3p, and induces EMT via TGFβ2 in A549 cells

PONE-D-24-11976R1

Dear Dr. Munekazu Yamakuchi, 

We’re pleased to inform you that your manuscript has been judged scientifically suitable for publication and will be formally accepted for publication once it meets all outstanding technical requirements.

Kind regards,

Arturo Aguilar-Rojas

Academic Editor

PLOS ONE

Reviewers' comments:

Reviewer's Responses to Questions

**Comments to the Author**

1. If the authors have adequately addressed your comments raised in a previous round of review and you feel that this manuscript is now acceptable for publication, you may indicate that here to bypass the “Comments to the Author” section, enter your conflict of interest statement in the “Confidential to Editor” section, and submit your "Accept" recommendation.

Reviewer #1: All comments have been addressed

Reviewer #2: All comments have been addressed

2. Is the manuscript technically sound, and do the data support the conclusions?

Reviewer #1: Yes

Reviewer #2: Yes

3. Has the statistical analysis been performed appropriately and rigorously? 

Reviewer #1: Yes

Reviewer #2: Yes

4. Have the authors made all data underlying the findings in their manuscript fully available?

Reviewer #1: Yes

Reviewer #2: Yes

5. Is the manuscript presented in an intelligible fashion and written in standard English?

Reviewer #1: Yes

Reviewer #2: Yes

6. Review Comments to the Author

Reviewer #1: The authors have addressed my concerns with clarifying language. As it stands, the article is structurally sound, but significantly limited in scope. Without additional experiments to address the questions raised by myself and by reviewer #2, the study allows only a very conservative interpretation of the data to infer involvement or not-194-3p in acidity-induced EMT-like changes in the studied cell type.

Reviewer #2: I sincerely thank the authors for thoroughly and thoughtfully addressing all the comments and questions raised in the previous review. The responses provided are clear, well-founded, and demonstrate a solid understanding of the subject matter, effectively addressing the observations made. The revised manuscript now meets the expected level of scientific rigor and quality, making a significant contribution to the body of knowledge in this field. In my opinion, the article is ready to be accepted and published in its current form, as it presents robust, novel, and relevant content for the scientific community. I commend the authors for this important work.

7. PLOS authors have the option to publish the peer review history of their article (what does this mean? ). If published, this will include your full peer review and any attached files.

**Do you want your identity to be public for this peer review?** For information about this choice, including consent withdrawal, please see our Privacy Policy .

Reviewer #1: No

Reviewer #2: **Yes: ** Mauricio Flores-Fortis

---

## [Editor Report · Acceptance letter]

PONE-D-24-11976R1

PLOS ONE

Dear Dr. Yamakuchi,

I'm pleased to inform you that your manuscript has been deemed suitable for publication in PLOS ONE. Congratulations! Your manuscript is now being handed over to our production team.

Kind regards,

on behalf of

Dr. Arturo Aguilar-Rojas

Academic Editor

PLOS ONE